# A Joint Sedimentation-Flood Retention Assessment of a Small Water Reservoir in Slovakia: A New Hope for Old Reservoirs?

**Peter Valent \*, Roman Výleta and Michaela Danáčová**

Department of Land and Water Resources Management, Faculty of Civil Engineering, Slovak University of Technology in Bratislava, Radlinskeho 11, 810 05 Bratislava, Slovakia; roman.vyleta@stuba.sk (R.V.); michaela.danacova@stuba.sk (M.D.)

\* Correspondence: peter_valent@stuba.sk; Tel.: +421-2-59-274-727

**Abstract:** The intensification of agricultural production brings problems related to water erosion, even to the upper parts of river basins. Soil particles that have eroded from unprotected agricultural land are often deposited in small water reservoirs, the efficiency or function of which might be compromised. This study presents an analysis of changes in the retention capacity of a small water reservoir over a period of 8 years. Within the study, a detailed bathymetry of the reservoir was conducted using an acoustic Doppler current profiler. The results, when compared to a 2008 geodetic survey, showed that the retention volume of the reservoir was reduced by only 2%, which was also confirmed by mathematical modeling. The possibility of strengthening the reservoir's role in flood protection was also investigated. A flood wave with a return period of 100 years was estimated using a design storm approach. A simple numerical model was proposed to transform the flood wave through the reservoir by considering four different scenarios of the elevation of the initial water level. The model, which is based on a water balance equation, uses simple hydraulic relationships to control the discharge through the reservoir's outflow objects. The results demonstrate that by reducing the initial water level, significant improvements in terms of the flood peak's attenuation and a longer time to peak values could be achieved.

**Keywords:** reservoir sedimentation; Universal Soil Loss Equation (USLE); sediment delivery ratio (SDR); acoustic Doppler current profiler (ADCP); flood wave transformation

## 1. Introduction

Water and water resources are often characterized by a high degree of spatial and temporal variability in both their qualitative and quantitative domains [1]. This variability is mostly determined by many physiographic factors such as climate, geology, soil, topography, and vegetation. Even though the demand for water is highly variable, it does not overlap with the naturally available supplies, which potentially results in multiple economic and material losses in times when it is both deficient or excessive. Over the centuries, it has been shown that the most effective way to balance the inequalities between the demand for and supply of water is to store it in reservoirs when it is abundant and release it when it is needed. Water reservoirs play an irreplaceable role in the integrated water resources management of a catchment, where they have become an important landscape-forming element. They can fulfil a large number of functions, among which the most important ones are, e.g., irrigation, water supplies, power generation, flood control, navigation, recreation, and fisheries. Apart from large multipurpose reservoirs, small water reservoirs (SWR) play an important role on a local scale; they often mitigate soil erosion processes and enhance the landscape-ecological conditions in the surrounding area [2,3].

The sedimentation of water reservoirs is recognised as one of the most significant threats responsible for the loss of available reservoir storage and for shortening the time of their lifespan [4]. Sedimentation is a natural process and cannot be fully avoided. Sediments, which are transported by a river flow either as a suspension or a bed load, are deposited in a reservoir due to reduced flow velocity and the associated decrease in its transport capacity. Coarse particles are always deposited first, with most of the reservoirs trapping 100% of the gravel and only small reservoirs that are situated in the upstream parts of watersheds being able to pass sand particles [4,5]. The main sources of sediments originate from riverbeds, eroded banks, and most notably hillslope surfaces in watersheds, from which the highest rates are generated on bare and agriculturally cultivated land. The consequences of sedimentation processes are far-reaching, with severe impacts on the reliability of water supplies, hydropower, reservoir infrastructures, flood control efficiency, and even a reduction of the recreational and biological potential of an area. Moreover, reservoir sedimentation also has significant impacts on the environment. As nutrients are mostly bound to fine sediments, their trapping in reservoirs could decrease the productive capacity of a river downstream of a dam [6]. Water which is deprived of the sediments is also referred to as "hungry water". It causes significant environmental damage to downstream river reaches, which are subject to severe water erosion processes that often lead to the deepening of a river channel and destruction of riparian ecosystems [5,7].

The rate at which a reservoir loses its useful volume is individual and depends on multiple factors. However, it has been reported that worldwide mean reservoir storage loss due to sedimentation ranges between 0.5 and 1% annually [8,9]. Once the storage of a reservoir is lost due to sedimentation, it is very difficult and expensive to reclaim it. Moreover, as good sites that are available for the construction of new dams are limited and the ability of water managers to enforce more efficient "grey" measures before "green" ones is getting more and more difficult, the adoption of a life-cycle management approach with a sustainable management of sediments should be encouraged [4,10]. The extensive research in reservoir management and the problem with its siltation resulted in a wide range of various techniques that can be used to protect reservoir storage from sedimentation. Annandale et al. [11] categorized the existing techniques as falling into three broad groups: (1) methods to route the sediments through or around a reservoir, (2) methods to remove sediments from a reservoir, and (3) approaches to reduce the sediment yield from an upstream watershed. A proper sustainable sediment management strategy should implement multiple techniques, with the greatest emphasis placed on enabling the sediments to pass the reservoir and preventing the detrimental impacts of "hungry water" further downstream [4]. Multiple studies have proved the efficiency of this approach with reservoirs in Japan and Puerto Rico bypassing around 87% and 95% of their total sediment loads respectively [12,13].

To adopt a suitable sediment management strategy, it is necessary to correctly determine the anticipated sediment yield from a river at the reservoir outlet and its expected sediment trap efficiency, which represents the percentage of the incoming sediments retained in the reservoir. Such information is of great importance not only in the design but also in the operation of existing reservoirs, where it can be used to optimize their operational manuals in an effort to improve their efficiency and reliability in fulfilling their functions (e.g., water supply, flood protection, hydropower). In general, the volume of sediments deposited in a reservoir over a specific period can be estimated either by calculating a difference between two direct measurements of the bed level conducted in different time intervals or by means of mathematical modeling. Nowadays, echosounders and other electronic devices enable measuring the bed level during the full operation of water reservoirs (see, e.g., [14–16]). However, as this process is rather time consuming and expensive, especially in the case of large reservoirs, it is often replaced by modeling the relationship between the soil erosion from the watershed and the sediment yield at the reservoir outlet. Despite the fact that a large number of various modeling approaches have already been developed [17,18], many of them utilize the concept of the sediment delivery ratio (SDR), which represents the ratio between the amount of sediment yield at river's cross-section and the soil erosion generated in upstream watershed. Zhang et al. [19] emphasized the importance of spatial

(watershed area) and temporal scales when estimating SDR. Once the sediment yield is estimated, only a fraction is deposited in a reservoir. This fraction is represented by the reservoir's trap efficiency, which is correlated to a number of reservoir and watershed characteristics and could be estimated using empirical relationships such as, e.g., Brown, Brune, or Churchill curves [20–22].

According to a review conducted by Jurík et al. [23], there are currently over 300 small water reservoirs registered in Slovakia. Most of them were constructed in the second half of the 20th century when massive investments into a large number of different water management projects were planned. The purpose of the majority of the SWRs was to provide water for existing or planned irrigation projects, of which only a few are currently still in operation. Since 1989, all SWRs have been administered by the Slovak Water Management Enterprise (SWME), the primary objective of which concerning the SWRs is to ensure that all of the reservoirs' structures are fully operational and capable of safely conveying a flood wave with a return period of 100 years. Nowadays, the still existing SWRs are predominantly used for recreational purposes and fisheries, which add only marginally to their current economic value, apart from that represented by the ecosystem services they provide [24]. The poor exploitation of the SWRs in fulfilling the SWME's tasks has resulted in a lack of financing dedicated to their maintenance and repair. This has resulted in the critical state of their outlet structures, including emergency spillways, which pose a threat to the areas downstream of the reservoirs. Apart from these problems, sedimentation from nearby agricultural land has been identified as the main factor responsible for their operation being inefficient and uneconomical at present. Šoltísová [2] examined the state of SWRs in Eastern Slovakia and concluded that in some cases, more than 60% of their capacity has been lost due to sedimentation (e.g., the Hrčeľ and Kľušov SWRs). Currently, the SWME is considering the possibility of including some of the SWRs into the existing flood protection schemes by utilizing a fraction of their volume to create a retention volume that would store flood waters. To do so, it is important to evaluate the current state and dynamics of the reservoirs' sedimentation rates. Moreover, it is necessary to conduct a subsequent analysis of their economic efficiency when restoring their original functions (in most cases, supplying water for agriculture) or their use in reducing flood risks and mitigating their impacts. The process of evaluating reservoirs must also include recalculations of design flood waves and reassessments of the reservoirs' characteristics and the security of the emergency spillways, as the data used for their design were of low quality due to the limited number of observations available [25].

This work assesses the possibilities of incorporating an existing SWR situated near the village of Vrbovce SWR, which is currently only used as a fishery, into its flood protection scheme. The main objective of this work was to estimate the sedimentation rate of the SWR by: (1) direct monitoring of the bed level using an acoustic Doppler current profiler (ADCP), and (2) mathematical modeling of the sediment yield and the reservoir's trap efficiency. Besides, a 100-year flood wave was estimated using a design storm approach, and a simple water balance model of the reservoir was proposed to model its transformation effect. Due to the need to preserve the fishery function of the reservoir, four different scenarios of the available retention volumes were considered.

## 2. Materials and Methods

### 2.1. Study Area

The study was conducted in a small area near the village of Vrbovce in the western part of Slovakia (Figure 1). The area lies in a hill-slope dominated country of the White Carpathians mountain range, with an altitude ranging between 340 m a.s.l and 580 m a.s.l. In the past, the whole area of the Myjava district underwent a significant and rapid transformation from a naturally forested to an agricultural landscape [26]. The typical character of the landscape in the first half of the 20th century was characterized by a mosaic of small and narrow plots that were often divided by vegetation strips, drainage furrows, terraces, or unpaved communications, which acted like significant surface runoff and erosion alleviation elements [26–28]. The communist collectivization of the agricultural land,

which took place after the end of World War II, resulted in the loss of this stable landscape, which was replaced by a number of large cooperative fields. This transformation led to an increased occurrence of flash floods and significantly intensified erosion processes, which resulted in the neighboring Myjava Hills becoming a part of the area with the highest density of permanent gullies in Slovakia [29]. The area belongs to a moderately warm climate region, with a mean annual air temperature of 7.5 °C and mean annual precipitation totals ranging between 700–800 mm. The bedrock is primarily composed of flysch-like rocks covered by loamy soils that create thick diluvial deposits (10–15 m) on the foothills and bottoms of dry valleys [29]. Since the beginning of the 21st century, the natural vegetation of oak-hornbeam forests can only be found on a small number of dispersed islands of little importance.

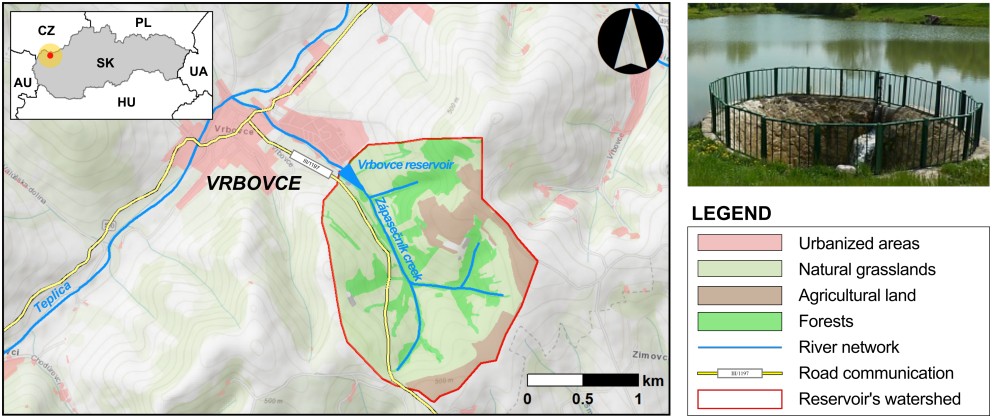

**Figure 1.** Location of the area of interest within Slovakia. The upper right-hand picture shows the emergency spillway with an embedded sluice gate acting as an additional outlet structure.

In the second half of the 20th century, the increased demand for irrigation water resulted in the building of a small water reservoir, which is situated on the Zápasečník Creek in the southeastern part of the village of Vrbovce (Figure 1). The reservoir, which was built in 1966, was originally designed for water supply purposes, but since the end of 1989 has predominantly been used for recreation and as a fishery. The watershed above the reservoir has an area of 3.24 km$^2$ and a mean altitude of around 450 m a.s.l. The higher altitudes and relatively steep slopes of the watershed, together with the reduction in agricultural production in Vrbovce, mean that, at present, only 17% of the watershed area is used for agricultural production. Figure 1 also indicates that all of the land used for agriculture is situated in the upper part of the watershed, which is characterized by mild values of slope gradients. Most of the watershed is covered by natural grasslands (56%), which are followed by small patches of broadleaf forest that constitute 25% of its area. The rest of the watershed is covered by artificial surfaces, transitional woodland shrubs, and water surfaces (2%).

The embankment dam of the reservoir has a height of 5.9 m, with a 3 m wide crest at an altitude of 350.0 m a.s.l. Its upstream slope has a ratio of 1:3 and is reinforced with a rip-rap, while the downstream slope has a ratio of 1:2 and is covered with grass. The bottom outlet from the reservoir is a concrete pipe of 150 mm in diameter, with its bottom edge lying on the reservoir's bed at an altitude of 344.1 m a.s.l. According to the current operational manual of the reservoir, this outlet is always kept closed. The water level in the reservoir can be partially controlled by a small vertical sluice gate (0.8 m × 1.0 m) embedded in the emergency spillway (see Figure 2). The emergency spillway was designed as an uncontrolled bell-mouth spillway, with its edge at an altitude of 348.25 m a.s.l. and an outer diameter of 6 m. The relationship between the elevation of the water level and the capacity of the reservoir was re-evaluated in 2017 (see Figure 3), during a bathymetry survey using an ADCP (Figure 4). Based on these measurements, the maximum volume of the reservoir, which is considered to be a water level 1 m above the edge of the emergency spillway (0.75 m under the dam crest), is more than 27,000 m$^3$ (Figure 2).

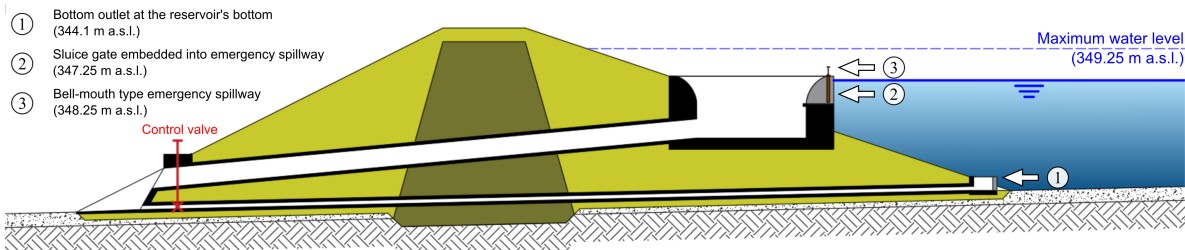

**Figure 2.** A schematic cross-section of the embankment dam of the Vrbovce small water reservoirs (SWR).

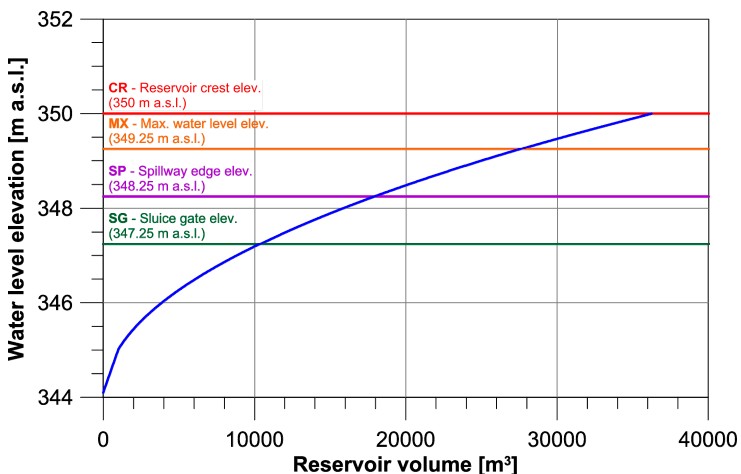

**Figure 3.** Elevation-capacity curve of the Vrbovce SWR.

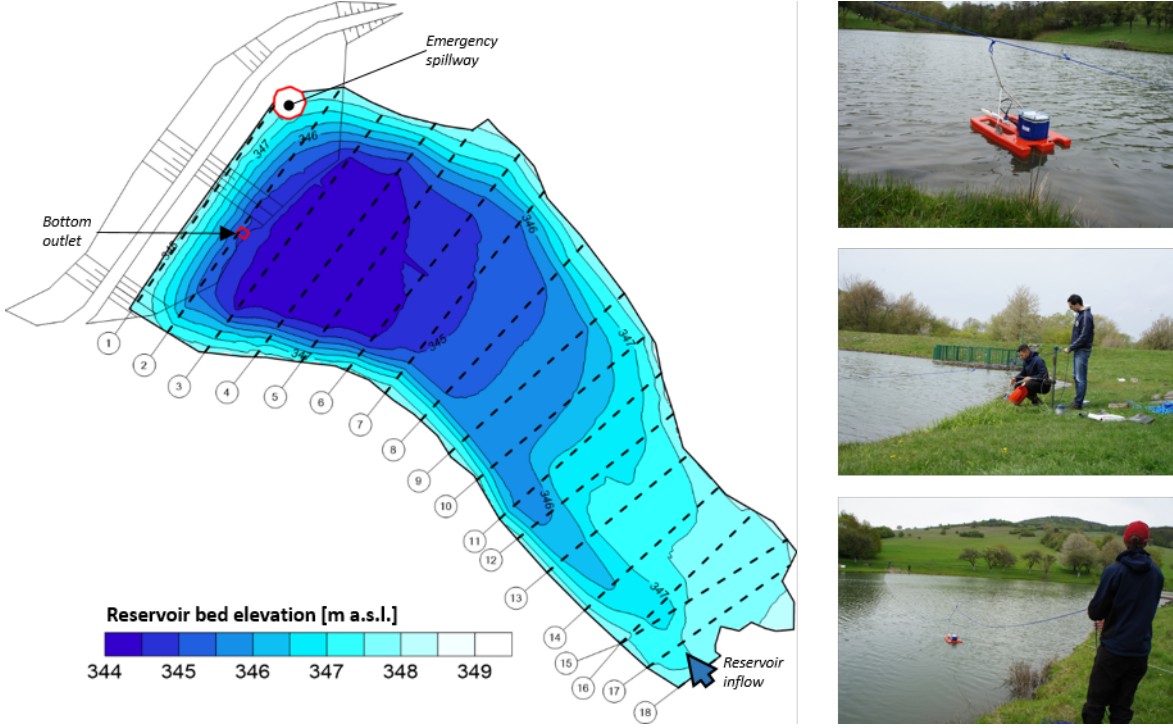

**Figure 4. Left**: Position of the 18 transects delineating the path of the acoustic Doppler current profiler (ADCP) measurements. Two measurements were carried out within each transect. **Right**: The 2017 bathymetry survey was based on towing the ADCP from one bank of the reservoir to another.

## 2.2. Bathymetric Survey of the Small Water Reservoir

In April 2017, a bathymetric survey of the Vrbovce reservoir was carried out. As the reservoir was in full operation during the survey, a method utilizing an ADCP was selected. The relatively small dimensions of the reservoir (max. width of only 65 m) permitted the use of Teledyne RD Instruments' StreamPro ADCP [30]. The transducers of the instrument are mounted on a small float, making it suitable even in shallow waters. The StreamPro ADCP has a working frequency of 2.0 MHz, with four transducers arranged in a Janus configuration oriented 20 degrees from the vertical axis. It enables the measurement of water depths in a range between 0.1 m to 7 m (assuming freshwater conditions) with an accuracy of 1%. The four piezoelectric transducers transmit and receive ultrasonic waves to measure the time needed for the waves to travel from a known point (the transducer's face) to a reflecting surface (the reservoir bed) and back. This, together with the known velocity of sound in water, enables calculating the depth of the water. The ADCP automatically adjusts the velocity of the sound in water as a function of its temperature, which is measured by a sensor mounted on the face of the transducer. As the maximum depth of the reservoir is rather small (~4 m) and as its thermal stratification was minimal during the April survey, no further adjustments to the velocity of the sound in the water were considered (see, e.g., [31,32]).

Given the relatively short length of the reservoir of only 150 m, 18 transects spaced regularly in one direction were used (Figure 4). Within each transect, a steel wire rope was stretched from one bank to another to guide the float and to prevent its deviation from the predefined course due to the effect of wind. Subsequently, the float was attached to the guide rope and manually towed to the other bank of the reservoir. In order to increase the reliability of the survey, two measurements were carried out within each transect. As the ADCP used was not equipped with a GPS sensor, the exact position of both of the transect's ends had to be recorded using a GNSS RTK system. The sampling frequency of the ADCP was set to 0.5 m since a higher level of detail would not significantly increase the precision of the 3D model of the reservoir's bottom. The obtained points were georeferenced and by using a simple triangulated irregular network (TIN) interpolation converted into digital elevation model (DEM) with a resolution of 0.5 m × 0.5 m. The reservoir's bed elevation was computed by subtracting the water depths measured from the elevation of the water surface, which was at 348.1 m a.s.l. at the time of the survey. To estimate the elevation-capacity curve of the reservoir, a detailed geodetic survey of the surrounding terrain was also conducted.

## 2.3. Modeling the Sedimentation in the Reservoir

Apart from the direct terrain measurements, the sedimentation of the SWR was also estimated using mathematical modeling methods. The modeling approach used was based on the relationship between the sediment yield entering the reservoir and the extent of soil erosion in the watershed. The mean annual soil loss from the watershed was estimated using an empirical model based on the Universal Soil Loss Equation (USLE). The method was developed in the USA for planar surfaces [33]; however, due to its simplicity and small data requirements, it is commonly used to quantify soil erosion from watersheds [34]. In its simplest form, it is given by the following equation:

$$A = RKLSCP \tag{1}$$

where *A* (t/ha/yr) is the average annual soil loss over the area of the hillslope; *R* (MJ cm/hour/ha/year) is the rainfall erosivity factor; *K* (t hour/MJ/cm) is the soil erodibility factor; *L* (-) is the slope length factor; *S* (-) is the slope steepness factor; *C* (-) is the cropping factor; and *P* (-) is the conservation practices factor. In this work, the soil erosion from the watershed was estimated using a GIS implementation of this method. The model enabled an evaluation of the individual factors and the resulting soil loss in a spatially distributed way in the form of raster maps with each cell covering an area of 10 m × 10 m. The value of the rainfall erosivity factor *R* was set by Alena [35] for the area of Western Slovakia at 28 MJ cm/hour/ha/year. The soil erodibility factor *K*, which represents the effect of soil properties on

erosion processes, was also processed by Alena [35], who employed the soil classification system used in Slovakia. The *L* and *S* factors were aggregated into a joint *LS* factor, which was calculated using an USLE2D model [36]. To reduce the uncertainties associated with the estimation of the aggregated *LS* factor, four different *LS* equations were used [33,37–39]. The final *LS* factor map, which was used to calculate the soil erosion, was determined by averaging the maps estimated by the four algorithms used. The value of the cropping factor *C* was determined for different types of land use based on the work of Malíšek [40]. The conservation practices factor *P*, the value of which is 1 for areas without any conservation practices, was only estimated for the agricultural land where contouring was used. As the efficiency of these practices decreases with the slope, a map of the DEM was used to account for this relationship [33].

It is important to realize that a certain amount of the eroded material estimated by the USLE is deposited within the watershed and thus is not transported to the watercourse and subsequently towards the reservoir. One way how to estimate the proportion of the soil erosion that ends up in the watershed outlet is to use a scaling factor, namely the SDR. The SDR is a ratio between the sediment yield and soil erosion over the same area and indicates the efficiency of the sediment transport of hillslopes and channel networks [19]. Several studies have shown that the SDR is related to a number of watershed characteristics including, e.g., the watershed area and shape, mean elevation, land use, soil type, stream network characteristics, climate, and annual runoff [17–19]. In the physical-geographic conditions of Slovakia, an empirical relationship derived by Williams [41] is usually used to estimate the SDR [42]. The relationship is given by:

$$SDR = 1.366 \times 10^{-11} \cdot F^{-0.0998} \cdot (ZL)^{0.3629} \cdot (CN)^{5.444} \qquad (2)$$

where *SDR* (-) is the ratio between the sediment yield and soil erosion; *F* (km$^2$) is the watershed area; *ZL* (m/km) is the relief-length ratio [43]; and *CN* (-) is the long-term average SCS curve number [44]. The sediment yield at the watershed outlet is then estimated by multiplying the soil erosion by the SDR.

As the value of the sediment yield only represents how much sediment could flow into a reservoir, one must further estimate the percentage that is likely to be deposited in the reservoir and not transported further downstream through its outlets in the form of a suspended load. To do so, the empirical relationship of Brune's curve [21] was used to estimate the trap efficiency of a reservoir as a function of the ratio between the reservoir's capacity and the mean annual inflow (see Figure 5).

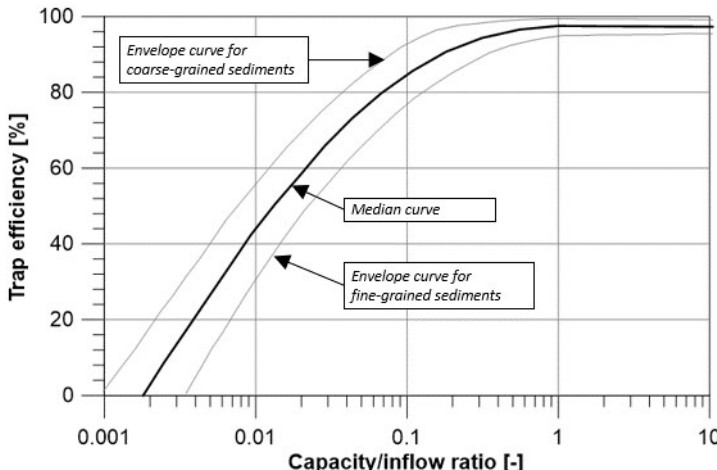

**Figure 5.** The relationship between a reservoir's trap efficiency and its capacity to the mean annual inflow runoff ratio as proposed by Brune [21].

One must note that even though this approach is very popular in estimating the sedimentation of water reservoirs, each of the methods used is a source of large uncertainties that may eventually

lead to erroneous results [17–19,34]. Therefore, the reservoir sedimentation rates estimated by the models presented should only have a complementary function to the monitoring of the terrain of the reservoir's sediment depth.

### 2.4. Modeling the Attenuation of a Flood Hydrograph

The effect of the small water reservoir on the attenuation of a 100-year flood wave was modeled using a simple numerical model programmed in the MatLab environment. The model is based on a simple water balance Equation (3), which simulates the outflow from the reservoir and the water volume increment at the end of the simulation time step. As the model was used to simulate the process of flood hydrograph attenuation, other members of the water balance, such as evaporation from a free water surface or losses through water seepage, were not considered. Moreover, the small area of the reservoir also enabled ignoring the routing of the flood wave through the reservoir and thus assuming that the inflow to a reservoir could immediately contribute to its outflow. The water balance used in the model is given by the following equation:

$$Q_{in}\Delta t - Q_{out}\Delta t = \pm \Delta V \tag{3}$$

where $Q_{in}$ (m$^3$/s) represents the discharge at the inlet to the reservoir; $Q_{out}$ (m$^3$/s) is the transformed output discharge; $\Delta t$ (s) is a time step of the simulation; and $\Delta V$ (m$^3$) is the difference in the volume. The time step of the simulation was set at 10 s to provide a sufficient degree of accuracy of the simulated variables. Larger values of the simulation time step would reduce the simulation time but on the other hand might compromise the reliability of the calculations during critical moments (spillway or dam overflows). The simulation is based on a known flood hydrograph, elevation-capacity curve, and the parameters of the outlet structures. It works in the following way: at the beginning of each time step, the inflow discharge is estimated from the flood hydrograph. The corresponding increase in the reservoir's water level is estimated from the elevation-capacity curve. Subsequently, the outflow from the first outlet structure (starts from the bottom) is calculated, and the new water level in the reservoir is estimated from the decreased water volume. In cases where the water level is higher than the bottom edge of the next outlet, the outflow from this structure is also calculated.

When applied to the Vrbovce SWR, the model simulates the output discharge as a combination of outflows from three different outlet structures. The first structure is a concrete pipe that lies on the bottom of the reservoir. It has a diameter of 150 mm and a slope of 2.3%. Due to its water supply and fishery functions, the small water reservoir is never fully emptied (prior to a flood event), which means that the pipe inlet is always fully submerged and that the pipe flow conditions are present. The model also assumes that the downstream side of the outlet pipe is free (not submerged) and that the capacity of the channel under the reservoir is large enough to convey even the most extreme discharges. The outflow from this structure is calculated as a pipe flow with a free outlet using the following formula:

$$Q_{out1} = S\sqrt{\frac{2gH}{1 + \sum \xi}} \tag{4}$$

where $Q_{out1}$ (m$^3$/s) is the discharge from the first outlet structure; $S$ (m$^2$) is the pipe's cross-sectional area; $H$ (m) is the energy head, which is defined as the difference between the water level elevation in the reservoir and the elevation of the pipe's axle at its outlet; and $\sum \xi$ (-) is the sum of the loss coefficients (friction, inlet, outlet). As the outlet of the pipe is free, the outlet loss coefficient ($\xi_{out}$) could be ignored and set to 0. For the inlet loss coefficient ($\xi_{in}$), a value of 0.5 was selected for the sharp-edged entrance to the pipe [45]. The value of the friction loss coefficient was estimated using the following formula:

$$\xi_{fr} = \frac{125n^2 L}{D^{\frac{4}{3}}} \tag{5}$$

where $\xi_{fr}$ (-) is the friction loss coefficient; $n$ (-) is the Manning's coefficient of roughness (0.013 for a concrete pipe); $L$ (m) is the length of the pipe; and $D$ (m) is its diameter.

The second outlet structure is a small vertical sluice gate, which is embedded in the bell-mouth emergency spillway (see Figures 1 and 2). The "opening" in the emergency spillway, which includes the sluice gate, is 0.8 m wide and 1.0 m high. The sluice gate itself is made of wooden planks and can be manually adjusted to manipulate the water level in the reservoir. The outflow from this structure occurs when the water level elevation in the reservoir exceeds 347.25 m a.s.l. In the event the water level does not reach the bottom edge of the sluice gate, the outflow from this structure is calculated as an outflow through a rectangular broad crested weir using the following formula:

$$Q_{out2,a} = \frac{2}{3}\mu_s b \sqrt{2gh}h^{\frac{3}{2}} \tag{6}$$

where $Q_{out2,a}$ (m³/s) is the discharge from the second outlet structure; $\mu_s$ (-) is a weir coefficient of 0.48 for a sharp inflow edge; $b$ (m) is the width of the outlet structure (0.8 m); and $h$ (m) is the height of the water, which is measured from the bottom edge of the outlet structure. In the event the water level exceeds the bottom edge of the sluice gate, the outflow is calculated as an outflow under the sluice gate using the following formula:

$$Q_{out2,b} = \mu_{sg}a_h b \sqrt{2g(h - h_c)} \tag{7}$$

where $Q_{out2,b}$ (m³/s) is the discharge from the second outlet structure; $\mu_{sg}$ (-) is the sluice gate contraction coefficient; $a_h$ (m) is the height of the sluice gate's opening; $b$ (m) is the width of the outlet structure (0.8 m); $h$ (m) is the upstream height of the water measured from the bottom edge of the outlet structure; and $h_c$ (m) is the minimum downstream height of the water ($h_c = a_h \cdot \varepsilon$). The contraction coefficient can be estimated as the product of the inflow velocity coefficient ($\varphi$) and the coefficient of vertical contraction ($\varepsilon$). In the event of free flow conditions (the outflow is not fully submerged), the value of $\varphi$ was set at 0.95 [46]. The value of $\varepsilon$ was estimated at every time step of the simulation as a function of the $a_h/h$ ratio [47] (Figure 6).

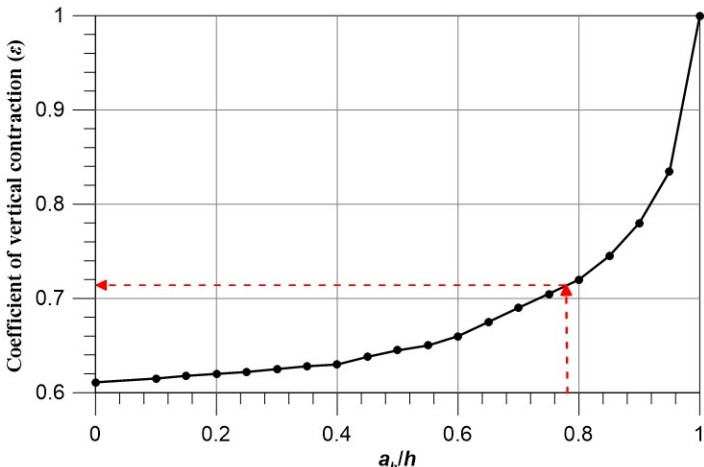

**Figure 6.** Estimation of the coefficient of vertical contraction $\varepsilon$ as a function of the $a_h/h$ ratio [46].

The last outlet structure is a bell-mouth emergency spillway (also known as a shaft, glory hole, or morning glory spillway), the crest of which is situated at 348.25 m a.s.l. The impact of the second outlet structure (the vertical sluice gate), which is embedded in the spillway (Figure 2), was not taken into account, as it would create a combined hydraulic structure, the parameters of which would have

to be estimated by experimental measurements. The model distinguishes between weir and orifice flow conditions [48], which are based on the value of the $h/D$ ratio. The following formula was used:

$$Q_{out3} = \begin{cases} \mu_{spill}l\sqrt{2g}h^{\frac{3}{2}} & h/D \leq 0.225 \\ \mu_{orif}S\sqrt{2gH} & h/D > 0.225 \end{cases} \tag{8}$$

where $Q_{out3}$ (m$^3$/s) is the discharge from the third outlet structure; $\mu_{spill}$ (-)is the spillway coefficient for the weir flow conditions; $l$ (m) is the length of the spillway crest; $h$ (m) is the height of the water measured from the spillway crest; $D$ (m) is the inner diameter of the spillway shaft (4 m); $\mu_{orif}$ (-) is the orifice coefficient for the orifice flow conditions (0.8) [47]; $S$ (m$^2$) is the cross-sectional area of the spillway shaft; and $H$ (m) is the difference between the water level elevation and the elevation of the bottom of the spillway shaft (346 m a.s.l.). The spillway coefficient was estimated at every time step of the simulation as:

$$\mu_{spill} = 0.461(h/r)^{0.033} \tag{9}$$

where $r$ (m) is the radius of the spillway crest (3 m).

## 3. Results

The results of the work could be divided into two separate parts. The first part was dedicated to the sedimentation of the Vrbovce SWR. To estimate the reservoir's sedimentation rate two distinct approaches were used: (1) monitoring and (2) modeling. The second part of the work was focused on investigating the possibilities of utilizing the SWR in the flood protection of the village of Vrbovce. In order to do so, a 100-year flood wave was routed through the reservoir under various volume retention scenarios dedicated to flood protection.

### 3.1. Sedimentation of the Vrbovce SWR

When using a monitoring approach, the quantification of the reservoir's sedimentation is based on the direct measurements of its bed at two different periods. After constructing detailed bathymetries for each period, it is possible to obtain information about the depth of the sediments at each point of the reservoir's bed accumulated over a period between the two measurements. This is done by subtracting the more recent bathymetry from the older one. Within this work, the most recent monitoring was carried out in April 2017. It was carried out during the full operation of the SWR and utilized an ADCP for monitoring the water depths. A geodetic survey from June 2008 was used as the reference monitoring. This monitoring was carried out at the end of maintenance works that were focused on the reconstruction of the outlet's structures and the removal of the bed sediments by a dry excavation method. In the next step, the data from both measurements were used to create DEMs (resolution of 0.5 m × 0.5 m) of the reservoir's bed that correspond to a water level of 348.1 m a.s.l., which was present during the 2017 monitoring and is kept constant throughout the whole year. A map of the sediment depth, which identifies areas that are subject either to the deposition of sediments or erosion, is shown in Figure 7. It demonstrates that over a period of eight years, the depth of the sediments increased in most parts of the reservoir. The largest sediment deposits of almost 2.0 m could be found under the heel of the dam, close to the bottom outlet structure and near the reservoir's left bank. In this area, a trapezoidal channel was excavated at the bottom of the reservoir during the 2008 maintenance work to create a preferential path for the bed sediments towards the dam and the bottom outlet structure. Figure 7 also identifies places where the reservoir's bed was subject to erosion. Most of them are situated along the left and right banks of the reservoir, which indicates that these areas have been exposed to bank erosion caused either by their instability or the erosive effect of the water. The results showed that the total volume of the sediments that have accumulated in the reservoir during the period between the two monitorings was 334 m$^3$. Taking into account the fact that after finishing the maintenance work in 2008, the most significant part of the reservoir's volume was not refilled earlier than after snow melting in the spring of 2009, it is reasonable to assume that the amount of sediments

accumulated over the period of eight years (excluding the period between June 2008 and April 2009). The mean annual sedimentation rate in the reservoir was then estimated to be 41.7 m$^3$/year.

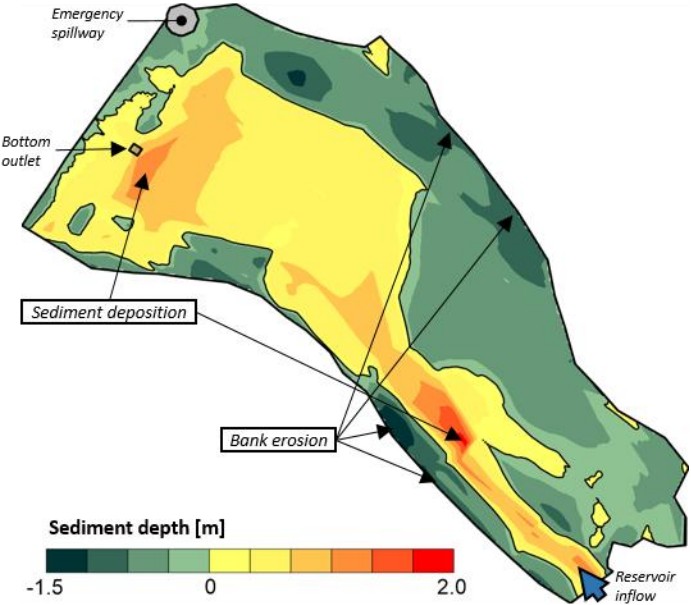

**Figure 7.** Map of the depth of the sediment of the Vrbovce SWR calculated by subtracting digital elevation models (DEMs) from the 2017 and 2008 surveys.

In the second approach, the sedimentation of the Vrbovce SWR was estimated using empirical models. In the first step, the amount of soil erosion from the whole watershed was modeled using a spatially distributed USLE model. As no significant changes in land use occurred between the two monitorings in the watershed, the simulations were carried out using a current land use map from 2017. The outcome of the modeling was a mean annual soil loss from the watershed of 0.995 t ha/year. In the next step, the SDR was estimated; it represented the ratio between the sediment yield and soil erosion. The value of the CN parameter used to calculate SDR was estimated as a mean value of the SCS curve number assigned to each cell of the rasterized catchment. The value of the estimated CN parameter was 66.2. The ZL parameter in Equation (2), which accounted for the shape and morphology of the watershed, was 41.7 m/km. The estimated value of the SDR was 0.385, which means that only 38.5% of the estimated mean annual soil loss from the watershed could be transported to its outlet, which was placed in the centre of the dam of the SWR. Furthermore, only a certain percentage of the sediment yield is deposited in a reservoir. The amount depends on the reservoir's trap efficiency, which was estimated using an empirical curve proposed by Brune (Figure 5), as a function of the ratio between the reservoir's capacity and its mean annual inflow. When estimating this ratio, the reservoir's capacity of 16,700 m$^3$ was used (the volume corresponding to the water level elevation of 348.1 m a.s.l.). Due to the fact that there is no water gauge station upstream of the reservoir, a hydrological analogy method was used to estimate the inflow to the reservoir. A nearby Myjava water gauge, which is situated in the neighboring catchment with identical climatic and geographic characteristics (distance to the reservoir of only 7.7 km, a watershed area of 32.02 km$^2$), was used for the analogy. The analysis of the mean daily discharges from the Myjava water gauge resulted in the estimation of a mean annual specific runoff of 8.8 l/s/km$^2$ (this corresponds to the value given for this region in a map of the mean annual specific runoff in Slovakia as shown in [49], p. 102). The value of the reservoir's capacity/inflow ratio was 0.019, which gives the range of the reservoir's trap efficiencies to be between 45.9% and 68.1% (a median of 56.9%), which was based on the size of the sediment particles. After accounting for the trap efficiencies, the resulting mean annual amount of sediments deposited in the reservoir was

between 51.1 m$^3$/year and 76.8 m$^3$/year, assuming their dry density of 1100 kg/m$^3$ [5]. The results of the modeling approach are summarized in Figure 8.

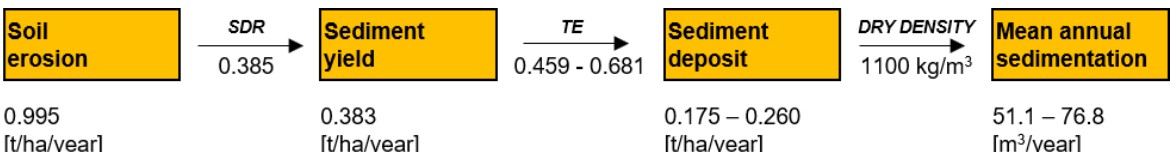

**Figure 8.** Summary of the modeling approach for the estimation of the reservoir's mean annual sedimentation rate.

### 3.2. Transformation of the 100-Year Flood Wave

Currently, the Vrbovce SWR does not serve its original function of supplying water for agriculture. Instead, it is being used as a fishery and for recreational purposes, which impose specific requirements on the water level during the year. To maintain favorable conditions for a number of fish species, the water level is kept constant throughout the whole year, i.e., just slightly under the edge of the emergency spillway at 348.1 m a.s.l. Under the current operational manual, the bottom outlet structure is not being operated at all and is kept permanently closed. This setting constitutes a retention volume that is available to store flood waters of only 1260 m$^3$, when considering the volume between the usual water level (348.1 m a.s.l.) and the edge of the emergency spillway (348.25 m a.s.l.). The retention volume increases to 10,940 m$^3$ when the maximum designed water level of 349.25 m a.s.l. is considered.

In order to assess the possibilities of using the SWR in the flood protection of the village of Vrbovce, four alternative scenarios were proposed to assess its efficiency in attenuating a flood wave with a return period of 100 years. The flood wave was estimated using a design storm approach under the assumption that a design storm event generates a flood wave of the same return period (see [50] for a thorough description of the method used). The selected shape of the flood wave hydrograph had a triangular shape with a rising to falling limb ration of 1:2 (see Figure 9a). This shape was selected based on an analysis of flood hydrographs observed at a nearby station in Myjava. The first scenario represents the current state of the reservoir's operation, with the initial water level at 348.1 m a.s.l. and a closed bottom outlet. The other three scenarios take into account the impact of the lowered initial water level on the reservoir's transformational effect. To preserve the current functions of the SWR, the maximum difference between the current and proposed water level elevations was only 1.5 m. The adjustment of the water level could be achieved by opening the bottom outlet prior to the expected flood event to prolong the time before the water level reaches the edge of the emergency spillway. Moreover, the outlet would be open throughout the whole flood event. The results of the flood wave transformation, together with the initial water level elevations of the individual scenarios, are displayed in Table 1. Figure 9 shows the simulated hydrographs of the transformed 100-year flood wave and the time course of the corresponding water level elevations. In all of the scenarios, the water level in the reservoir exceeds the edge of the emergency spillway. The simulations revealed that while a transformation effect (calculated as ($Q_{in,peak}$-$Q_{out,peak}$)/$Q_{in,peak}$*$100$)) of only 8.1% was achieved in the case of scenario 1, efficiencies from 20.3% to as much as 52.5% were obtained in the three other alternative scenarios (Table 1). Moreover, significant improvements were achieved in terms of the delay times (the duration between the flood and transformed peak flows), which rose from only 7.5 min (scenario 1) to as much as 49.0 min (scenario 4).

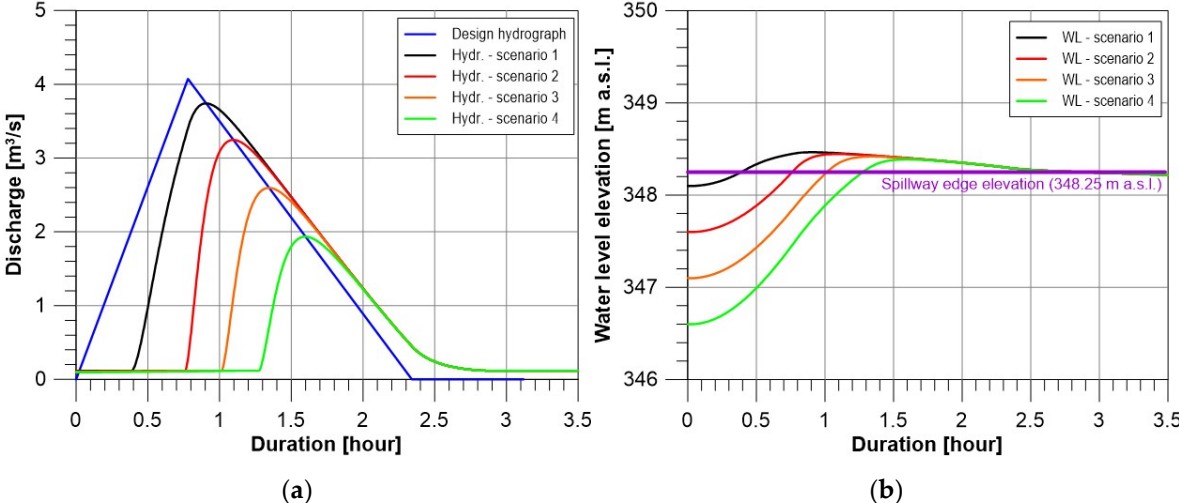

**Figure 9.** (**a**) Hydrographs of the transformed 100-year flood waves; (**b**) and the corresponding water level elevations.

**Table 1.** Summary of modeling the transformation of a 100-year flood wave through the Vrbovce SWR. Four scenarios with different initial water level elevations were considered.

| | Initial. WL (m a.s.l.) | $Q_{out,max}$ (m³/s) | Max WL (m a.s.l) | Transformation Effect (%) | Delay (min) |
|---|---|---|---|---|---|
| Scenario 1 | 348.1 | 3.74 | 348.46 | 8.1 | 7.5 |
| Scenario 2 | 347.6 | 3.25 | 348.45 | 20.3 | 18.8 |
| Scenario 3 | 347.1 | 2.59 | 438.42 | 36.2 | 33.8 |
| Scenario 4 | 346.6 | 1.93 | 348.39 | 52.5 | 49.0 |

## 4. Discussion

### 4.1. Monitoring Approach and Its Uncertainties

When comparing the methods used to estimate the sedimentation of the Vrbovce SWR, a method based on the monitoring of water depths using an ultrasonic ADCP was used as the reference method. On the one hand, the advantage of this method lies in the precise measurement of the reservoir's bed elevation, which is given by the parameters of the device used. In ideal conditions, the measurement error may often be less than 1%. However, it rises with the increasing turbidity of the water and its depths. This is often accompanied by significant thermal stratification, especially during the summer and winter months [31,51]. This problem has also been addressed in several studies that have shown that a 5 °C error in temperature measurements throughout an entire water column could bias the water depth measurements by approximately 2% [31]. In the case of the bathymetric survey at the Vrbovce SWR, no circumstances that would significantly bias the measurements could be observed. Another advantage of this method is the speed of the monitoring, which permits the collection of large amounts of data in a relatively short period of time. A sufficient spatial density of the measurements is essential for creating a reliable model of the reservoir's bed and a subsequent unbiased estimation of its sedimentation rate. On the other hand, one must also be aware of the disadvantages associated with this method. One of them is the need for a second measurement to determine the thickness of the sediments at a particular place and to estimate the mean annual sedimentation rate of the reservoir. The quality and accuracy of this reference monitoring directly affect the reliability of the estimated reservoir's sedimentation characteristics. In the case of the 2008 monitoring, which took place immediately after the dry excavation of the deposited sediments, the elevation of the reservoir's bed and the geometry of its banks were measured during a conventional geodetic survey. As the monitoring was primarily oriented towards fixing the position of the outlet

structures and the geometry of the dam itself, only a small number of measurements were used to survey the reservoir's bed. Despite this, the measurements were taken at locations with apparent changes in the bed's geometry, which made them adequate for the estimation of the sediment depths. Another disadvantage of this method is the relatively high cost of the equipment used. Moreover, when used for a larger reservoir, the ADCP must be attached to a boat, which further increases the cost of the monitoring. Such a setting also makes it impossible to measure closely in the vicinity of the reservoir's banks.

### 4.2. Modeling Approach and Its Uncertainties

In the event such a modeling approach is used, one must be aware of the various sources of uncertainties that can influence both the accuracy and reliability of the sedimentation estimates. The main sources of the uncertainties are associated with the quality and quantity of the input data (here, e.g., DEM, land use map, soil texture map, inflow to the reservoir), as well as with the selection and structure of the models used. When modeling the sedimentation of the reservoirs, multiple authors recommend not making a single number prediction, but to instead provide a range of potential results, or if possible, include error bars or a distribution of the potential results [17,52]. In this way, it is possible to account for the different sources of uncertainties that are an inevitable part of a modeling approach. The main advantage of modeling a reservoir's sedimentation, when compared to a monitoring approach, is the simplicity of the calculations used, the relatively small requirements for the input data, and the possibilities of predicting sedimentation rates for prospective reservoirs. Moreover, the modeling approach also enables accounting for the impact of a number of factors, such as land use or climate change, erosion mitigation measures, or new crop management practices on agricultural land.

In this study, each of the methods used is associated with a certain degree of uncertainty. When modeling the mean annual sedimentation rate as presented here, the individual methods are represented by USLE for the quantification of soil erosion, SDR for estimating its proportion that ends up in the reservoir, and the empirical Brune's curve for evaluating reservoir's trap efficiency. In case of the soil erosion model, some of the uncertainty comes directly from the type of the model used. Despite being widely used the USLE model was originally designed to estimate soil erosion from small agricultural plots. Because of this, it is not able to account for various landscape features that might act as a sediment transport barrier or simulate the complexities of rill and gully erosion processes. As a result, the modeller should be very careful when applying the model on small watersheds and use it only in cases when sheet erosion processes prevail. For other cases, more complex physically based models should be used. Another source of the uncertainty comes from the model parameters. In case of the USLE model the biggest problems are associated with the estimation of the rain erosivity parameter $R$. Some of the existing methods return values of $R$ that can differ by an order of a magnitude, which influences the USLE model estimates proportionally. The most reliable estimates of $R$ can be obtained by analysing long time series (50 years or more) of high-frequency (30 min or less) precipitation measurements over an area of interest. As such data was not available for this study, the rainfall erosivity factor $R$ was obtained from the study of Alena [35] who analysed precipitation data from 22 rain gauges in Slovakia and interpolated the estimated values of $R$ to the areas without observations. The value of $R$ used in this study was also compared to the values estimated in a large European study [53] (1 km grid) with which it coincided. Another source of uncertainty is related to the estimation of the joined $LS$ factor which accounts for the combined effect of slope length and steepness. In order to reduce this uncertainty, four different methods were used to identify possible outliers. The values of the mean areal estimates of the $LS$ factor ranged between 8.4 to 11.0 with a mean of 10.3 used in the estimation of soil erosion. The rest of the uncertainty associated with the model parameters came from the estimation of parameters $K$ (soil erodibility), $C$ (cropping factor) and $P$ (conservation practices factor). As the degree of freedom in estimating these parameters is relatively small and their values in

Equation (2) are negligible to those of parameters *R* and *LS*, they contribute only marginally to the total uncertainty associated with the USLE model.

In case of the SDR model, which was used to predict the amount of sediments that could end up in the catchment outlet, the potential sources of uncertainty were associated mainly with the structure of the model and the validity of the assumptions it was built with. Goudie [54] in his study states that a value of SDR can be anywhere between 0.3 to 1 for small catchments (around 0.1 km$^2$) and 0.02 to 0.2 for large catchments (1000 km$^2$). Even though several studies have shown a direct link between the observed values of SDR and selected catchment morphology characteristics, the exact value of SDR is always site-specific and not clear [55]. Post and Harcher [55] showed that only by moving from a constant to a spatially variable SDR model the amount of sediment yield entering river network doubled. Moreover, one should also consider the fact that the SDR values might be conceptually defined differently from one study to another [54], which makes it even more challenging to select the best model for a given catchment. To enable the intercomparison with different studies in Slovakia the method of Williams [41] was used as a method usually used in Slovakia for this type of studies [42]. As most of the parameters used in this method are directly derived from the catchment characteristics, the parameter uncertainty is minimal.

In the case of the model used to estimate the reservoir's trap efficiency the uncertainty originates from both the model structure and model parameters. The method of Brune [21], which was used in this study, uses an empirical relationship between reservoir characteristics (ratio between its capacity and mean annual inflow) and its trap efficiency while considering the size of the sediment particles. The largest portion of the uncertainty lies in the fact that the empirical relationship used in the method was derived for a different geographical region (USA). This means that one must be very careful when using the method outside the region for which it was derived and interpret the results only in the light of these facts. From among the inputs to the method the highest amount of uncertainty is associated with the size of the sediment particles and the mean annual inflow to the reservoir. Within this study, the size of the sediment particles, which determines the position of the empirical curve (see Figure 5), was estimated based on the laboratory analysis of soil samples (loamy soils). The estimation of the mean annual inflow to the reservoir could be slightly problematic especially in ungauged catchments. Even though this is also the case of this study, the close distance of a gauged watershed with identical morphological, climatic and land cover characteristics reduced the uncertainty associated with this input to a minimum.

It must be noted that the uncertainties as mentioned above might lead to either erroneous estimates of the mean annual sedimentation rates or the estimates with a large variance. The level of uncertainty could be reduced by incorporating local knowledge of model inputs. Moreover, the means of representing uncertainty could also be improved by placing confidence bounds on the predictions. However, to do so a better understanding of model parameters in terms of their distribution and limits is needed so more sophisticated approaches utilizing Monte-Carlo simulations could be applied.

## 4.3. Vrbovce SWR—Results and Implications

Table 2 summarizes the values of the mean annual sedimentation rates (MASR) of the Vrbovce SWR, which were estimated by both monitoring and modeling approaches. In the case of the modeling approach, a range of the possible estimates is given based on the reservoir's trap efficiencies, which were estimated from Brune's empirical curves for fine and coarse-grained sediments. The results indicate that, when compared to the results of the monitoring approach, the modeling overestimates the values of MASR by 22% in the case of fine-grained sediments and by as much as 84% in the case of coarse-grained sediments. However, when taking into account the fact that the soil texture in the watershed is mostly loamy or clayey-loamy (fine-grained soils), it is reasonable to consider the lower values of the MASR as the more representative ones. Nevertheless, the difference between the modeled and monitored values of the MASR could be caused by a number of different factors. One of them could be the uncertainty associated with estimating the soil erosion from the reservoir's

watershed. The USLE model used for this task was originally developed to predict soil erosion from flat surfaces used for agricultural production [33]. Moreover, as multiple studies have suggested, its parameterization is often a great source of uncertainty [56,57]. In this study, the agricultural land is situated only in the upper part of the watershed (Figure 1). It is therefore questionable to what extent the model is able to take into account the protective effect of the natural grasslands and forests. Both of these types of land cover are characterized by their ability to significantly slow down transport processes and thus reduce the amount of sediments entering a reservoir. Specifically, the position of the forests that are situated in the vicinity of the watercourse and which act as a significant erosion control measure is challenging to account for in the process of SDR estimation. Another factor that could explain the difference between the modeled and monitored value of MASR is the possible impact of the road communication III/1197, which is situated on the left side of the reservoir and which runs across the whole watershed (Figure 1). As a small trench is also a part of the road profile, it may act as a barrier that prevents sediments from entering the reservoir. Moreover, in the case of extreme precipitation events, the trench can create a preferential path that transports them further downstream the reservoir. To assess this hypothesis, further modeling was carried out in which the part of the watershed above the road communication was not included in the modeling. The estimated values of the MASR ranged between 37.5 m$^3$/year and 55.7 m$^3$/year, which represents a $-10\%$ and $+33\%$ difference when compared to the estimates using the monitoring approach. Such results correspond very well to those estimated using the monitoring approach, especially when narrowing the range of the estimated MASR values based on the expected size of the sediment particles (the reservoir's trap efficiency estimated from the region between Brune's median and bottom envelope curves). In such a case the estimated MASR value for the Vrbovce SWR would be very close to ~40 m$^3$/year.

**Table 2.** Summary of the estimation of the mean annual sedimentation rate of the Vrbovce SWR as estimated by monitoring and modeling approaches. The two modeling approaches represent two scenarios with and without taking into account parts of the watershed above the III/1197 road communication.

| Method for Estimating Reservoir's Sedimentation | TS (m$^3$) | MASR (m$^3$/year) | DIFF (%) |
|---|---|---|---|
| Monitoring | 334.0 | 41.7 | - |
| Modeling—with the area above the road | 408.8–614.1 | 51.1–76.8 | +22% to +84% |
| Modeling—without the area above the road | 300.0–445.6 | 37.5–55.7 | −10% to +33% |

Note: TS—total sedimentation over a period between the two bathymetry monitorings (8 years); MASR—mean annual sedimentation rate; DIFF—difference between the modeled and measured MASR.

Such a value, which was confirmed by both approaches, represents the very small sedimentation rate of the Vrbovce SWR. The main reason for this is the way in which the land is used and structured. In 2017, only 17% of the reservoir's watershed was used for agricultural production. Its position in the upper part of the watershed, together with large areas of natural grasslands and the banks of the watercourse protected by forests, are the most important factors responsible for this state. Under the current composition of the land use types in the watershed and the estimated value of the MASR of 41.7 m$^3$/year, only 2.3% of the reservoir's capacity (currently 18,051 m$^3$ when considering the edge of the emergency spillway) would be lost in 10 years due to sedimentation. Even though such a small rate of sedimentation does not require any new measures to be taken in the watershed, the current state of the outlet structures requires to improve their maintenance. The functioning of the bottom outlet structure is especially vulnerable since greater amounts of sediments tend to be deposited in its vicinity (Figure 7). Despite its small dimensions (only 150 mm in diameter), the structure could be kept open during periods of higher flows to flush away part of these sediments (see, e.g., [4]). It should be noted here that such a favorable situation regarding the sedimentation rate of the Vrbovce SWR is somewhat new. In the past, most of its watershed was used in agriculture, including the slopes around the reservoir. This resulted in a significant loss of the reservoir's capacity, which

would not be possible with the current rates of its sedimentation. In case the intensive agricultural production in the watershed was restored, adequate mitigation measures (e.g., proper choice of crops, crop rotation, conservation tillage, contour cultivation, ridging, and terracing) should be taken to reduce the increased amount of sediments entering the reservoir. These measures should prolong its lifespan.

The main task of the Slovak Water Management Enterprise (SWME), as the current administrator of the Vrbovce SWR, is the management of watercourses and the flood protection of urbanized areas [25]. At present, the Vrbovce SWR is not included in any flood protection scheme, which results in reduced interest from the SWME in its maintenance. Modeling the transformation of a 100-year flood wave through the reservoir showed that even a small decrease in the initial water level elevation could significantly reduce the peak flow and, in the case of the most extreme scenario 4, delay its occurrence by as much as 41.5 min when compared to the current state of its use, as represented by scenario 1 (Table 1). Despite the relatively small area of the watershed above the reservoir (only 3.24 km$^2$), its use in flood protection could reduce the flood risk not only in the village of Vrbovce, but also in areas further downstream. The fact that the reservoir does not serve its original function (and nothing suggests it will in the near future) favors the reclassification of its primary function to flood protection.

## 5. Conclusions

Nowadays at the end of the second decade of the 21st century, many small water reservoirs in Slovakia that were built before 1989 to provide water for irrigation and agricultural production do not serve their original functions. The lack of maintenance and the intensification of agricultural production in their watersheds has resulted in the main problem associated with their operation being excessive sedimentation from nearby hillslopes. The main objective of this work was to estimate the mean annual sedimentation rate of the Vrbovce SWR, which in 2008, underwent the reconstruction of its outlet structures and the mechanical removal of the sediments deposited. The monitoring of the bed sediments using the ADCP showed that the estimated mean annual sedimentation rate between the years 2008 and 2017 was only 41.7 m$^3$/year. This means that when considering the reservoir's capacity of 18,051 m$^3$, as given by the edge of the emergency spillway, only 2.3% of that capacity would be lost in 10 years. The results were also confirmed by modeling, which in addition identified the road communication III/1197 as a barrier that prevents eroded sediments from the part of the watershed above the road being transported to the reservoir. The possibilities of reclassifying the primary purpose of the reservoir, i.e., from a water supply to flood protection, was also investigated. To do so, a simple numerical water balance model was proposed to transform an estimated 100-year flood wave through the reservoir. The simulations revealed that despite the relatively small area of the watershed, the reservoir might play an important role in a system of flood control reservoirs while maintaining its current use as a fishery. At present, its incorporation into a flood protection system is one of the few ways to secure financing for its operation and maintenance and the preservation of its water management and ecosystem functions.

A similar situation as described here could be observed in most SWRs in Slovakia and other post-communist countries in Central Europe. Updating their actual capacities, sedimentation rates, abilities to fulfil their functions, and abilities to safely transform 100-year flood waves both under current and changed climatic conditions should only be the first steps to recover these critical water management structures.

**Author Contributions:** Investigation, R.V. and M.D.; formal analysis, R.V. and P.V.; methodology, P.V. and R.V.; software, P.V.; visualization, P.V.; writing—original draft, P.V.; writing—review and editing, P.V.

**Funding:** This research was funded by the Slovak Research and Development Agency, grant numbers APVV-15-0497 and APVV-15-0425; by the Slovak Science Grant Agency, grant number VEGA 1/0891/17; and by an international co-operation between the Czech and the Slovak Republics, contract number INTERREG V-A SK-CZ 304021C996.

**Conflicts of Interest:** The authors declare no conflict of interest.

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
