# Peer review of "A Joint Sedimentation-Flood Retention Assessment of a Small Water Reservoir in Slovakia: A New Hope for Old Reservoirs?"

_geosciences, doi:10.3390/geosciences9040158_

Round 1

Reviewer 1 Report

General comments:

The authors have presented an assessment of sediment accumulation in a small reservoir in Slovakia. They have used physical measurements, and developed a mathematical model based on their findings. The benefit of the measurements is that they show how fast the reservoir is filling with sediment since 2008 (not very fast). One benefit of the mathematical model is that it shows how a 100-year flood wave would be transformed through the reservoir. The techniques seem to be applicable for small water reservoirs in temperate zones. It does not seem that any new methodologies have been presented in this work, so I’d like to better understand how broadly applicable the findings are. Could the authors give a sense for the regions of the world in which sedimentation rates may be similar? Or am I doomed to find these results useful only if I’m working in Slovakia? Is there anything novel about the sediment measurement technique? Is there anything novel about the sediment modeling (Universal Soil Loss Equation) or flood attenuation? Did I miss it? This is a very important topic - sedimentation of reservoirs - and the authors have done something important by matching a model to physical measurements. It has potential to be very useful to readers and dam builders/operators. The writing is eloquent and the presentation is professional. But I request that the authors help me better understand 1) the novelty of the approach; 2) its regions of global applicability (applicability either of the methods or the results of this study); and 3) the uncertainty in / accuracy of the model.

Specific comments:

1.       The authors use the term “nowadays” a lot. Maybe it’s just my stylistic preference, but I find that word to be sloppy. Could you switch out that word for something more scientific? What are the bounds of “nowadays”? Within the last year? Within the 5 years? What transpired that transitioned us, in this case, into the epoch called “nowadays”? Try saying “since this event …” or “within the last 5 years” or “recent standard practice requires…”

2.       Line 400 – compares to a nearby gauge for a watershed that is 32.02 km2. But the study watershed is only 3.24 km2. 1/10th as big. I didn’t see where the correction was made to account for the vast difference in size. Is land use similar? And elevation range? Any snow differences?

3.       Line 489 – the authors say that the main advantage of modelling is predicting sedimentation for prospective reservoirs. That’s not my understanding of the main purpose of models. I think the main benefit of models is that they allow us to evaluate the response of the system to conditions outside of the single set of historical observations. They allow us to test a design to possible future conditions different from the past. To that point, have the authors evaluated the response of the reservoir sedimentation to conditions different from the past? What if climate change increases the 100-year flood? What if land use change results in more sediment? Can the model be used to evaluate these questions? If so, that would be of great interest, I believe.

4.       The discussion of Table 2 uncertainty is quite speculative. I would like to see it tested more analytically – a more systematic treatment of uncertainty. What, for example, is the uncertainty in the USLE? A model that could predict sediment accumulation in reservoirs with some measure of accuracy would be hugely useful. I am aware of the global heterogeneity in almost every contributing factor, and I sympathize with the challenges faced by the authors here, but that is exactly why I request that they look harder at where all the uncertainty comes from, so as to attempt to improve the accuracy of the tool. We, the global user community, could then be informed about the risks of applying their tool to our home watersheds.

5.       Table 2: the TS with and without the road seem quite dramatic. Could you provide some justification for the very large impact of the road on TS in the river? I’m surprised by its magnitude.

6.       Line 537 – even though small sedimentation rates, more maintenance required? Why? That statement seems unjustified.

7.       In equation 3, where are the vertical fluxes (evaporation and infiltration)? If the authors think they are negligible compared to the other terms, please provide justification.

8.       I teach fluid mechanics and hydraulics at my university, but I don’t know where all of the equations 4-9 come from. It would help me if there was a bit more derivation of those relationships. As of now, they feel thrown-down, and I’m not clear to what extent they are simply empirical, and to what extent they are universal.

Reviewer 2 Report

Dear Authors: 

It is an interesting research with erosion and sedimentation as a theme of research. 

Please address the comments on the PDF attachment. 

The reviewer

Reviewer 3 Report

The study of “Impact of soil erosion on the sedimentation of a small water reservoir: a flood protection perspective” is interesting that describes a case study about sediment and flood protection in Slovakia. Their survey supports the modeling result of sediment and investigates the flood protection of the small reservoir. Meanwhile, I have some comments below:

1.      The title cannot reflect the content of the manuscript. Its title needs to be modified to better represent its content, since this is a case study and does not investigate the impact of soil erosion on sedimentation of the reservoir. In addition, the research lacks novelty and new findings, such that it is not appropriate to use the current title. If there is something new, please add to the last paragraph of Section 1 to highlight your findings.

2.      As indicated in table 2, the modeling without the parts of road has better results according to the monitoring estimates. I suggest showing its results in Section 3.1 and Figure 8 instead of showing the results with road. Also revise the word “with road” “without road”, because it may confuse readers that it is better to model with the existing road. Did you mean modeling with/ without the parts of watershed above the road?

3.      English of the manuscript needs to be improved. For example, misuse the word “or” in lines 29, 52. Grammar errors in Lines 54 –55 (their becoming, water devoid), line 89 (at).

4.      Figure 3, could you explain why the blue curve has a higher elevation than reservoirs crest elevation?

5.      Use the correct format to represent the unit of variables. For install, do not use dot in the units, Line 223 and other places. Add the units of all variables in equations 3 – 9.

6.      For the LS in equation 1 lines 233 -234, do the four different LS vary significantly? How good if comparing your value to the work of developed product: A New European Slope Length and Steepness Factor (Geosciences 2015, 5(2), 117-126, https://www.mdpi.com/2076-3263/5/2/117/htm)

7.      What is the physical mean of ZL in Line 252? How do you determine CN curve number value as 66.2 in line 391? The assumed dry density of 1100 kgm-3 (Line 409) seems too low. The uncertainties of the parameters could lead to large variance of the estimated mean annual sedimentation, this can be discussed in Section 4.

8.      For the study of Vrbovce SWR, some data are using the data from Myjava stations (Lines 402 and 431), can this introduce some uncertainties?

9.      In line 547, “adequate mitigation measures should be taken to reduce the increased amount of sediments entering the reservoir.”, could you list some examples of mitigation measures?

Some other minor comments:

1.      Fully spell the words of “USLE; SDR; ADCP”

2.      Remove the last comma (,) in the equations.

3.      Revise “in order to” as “to”, because it is heavily used.

Round 2

Reviewer 3 Report

My comments are carefully addressed and well explained.